# Bridging the Gap: Integrating Theory and Practice for Early Childhood Physical Education Teacher Education in Korea

Sunghae Park [1]  and Hyunwoo Jung [2],*

1    Institute of Social Science, Gachon University, Seongnam 13120, Republic of Korea; psh8503@gachon.ac.kr
2    Department of Policy R & D, Korea Institute of Sport Science, Seoul 01794, Republic of Korea
*    Correspondence: hjung@kspo.or.kr

**Abstract:** Early childhood physical education (ECPE) plays a crucial role in giving young children their first impressions of physical activity and promoting lifelong participation. In the Korean context, however, the provision of physical education to young children is challenging owing to a lack of expertise among teachers. This study aims to explore effective teacher training strategies to normalize physical education for young children in Korea. Accordingly, the purpose of this study was to derive teacher education strategies that minimize the gap between theory and practice in ECPE. These integration strategies are presented in two categories: pre-service teacher training strategies and in-service teacher training strategies. Five sub-categories were identified within each category, goal, content, method, evaluation, and environment, and the educational implications in terms of minimizing the gap between theory and practice in ECPE teacher education were discussed based on the results. The teacher education strategies derived based on this study are significant in that they prioritize the integration of theory and practice based on the shared opinions of ECPE scholars and practitioners and offer concrete and practical approaches that can actively contribute to the promotion of ECPE.

**Keywords:** early childhood physical education; teacher education; integrating theory and practice

## 1. Introduction

Early childhood physical education (ECPE) for preschool-aged children holds significant importance in the overall development and well-being of young children. It plays a vital role in promoting motor development, physical health, cognitive abilities, social-emotional skills, and the establishment of lifelong physical activity habits [1]. Engaging in regular physical activity from an early age enhances motor skills, promotes healthy growth, improves cognitive function, fosters social interactions, and instills a love for physical activity. Research indicates the positive impact of ECPE on various domains of child development [2–4], paving the way toward lifelong physical and mental well-being by providing a foundation for healthy habits and holistic development.

However, ensuring high-quality physical education experiences for preschool-aged children is challenging in many countries [5,6]. Inadequate practice resulting from a lack of professional development among teachers is a significant problem in ECPE [7], as many teachers lack the necessary knowledge, skills, and training specific to physical education in preschool, resulting in suboptimal teaching practices and limited effectiveness in promoting children's physical development and overall well-being [8]. The absence of specialized training programs and professional development opportunities focused on ECPE exacerbates this issue, hindering teachers' ability to provide high-quality physical education experiences for young children. Research has emphasized the critical role of well-trained and knowledgeable teachers in delivering effective physical education programs in early childhood [9,10]. As such, it is crucial that the professionalism gap among ECPE teachers be addressed by means of a well-designed teacher education program, which

is crucial to ensuring that children receive the necessary guidance, support, and quality instruction for their physical development [11,12].

A similar problem exists in Korea, where ECPE for preschool-aged children has faced challenges because of a lack of professionalism among teachers [13,14]. The limited expertise and knowledge of teachers in ECPE hinder the implementation of effective teaching methods and curricula, impacting the quality of physical education for young children [15,16]. The issue arises from the absence of specialized training programs and professional development opportunities focused on ECPE, with the result that teachers lack the necessary skills and understanding of age-appropriate instruction. Put otherwise, the limitations that pre-service and in-service teacher education impose on the development of physical education expertise among preschool and kindergarten teachers in Korea significantly widen the gap between theory and practice by failing to effectively enable teachers to apply theory to their physical education classes or provide opportunities to generate knowledge in practical teaching settings [13–16].

The importance of providing effective teacher education in ECPE is widely recognized. Research suggests that teachers with appropriate training and knowledge in this domain are more likely to deliver developmentally appropriate activities, engage children effectively, and foster positive attitudes toward physical activity [8,17]. Conversely, inadequate teacher preparation can hinder the provision of quality physical education experiences, potentially leading to missed opportunities for children's physical, cognitive, and socio-emotional development [7,18]. In particular, exploring teacher education strategies that minimize the discrepancy between theory and practice is crucial for providing effective teacher education in ECPE.

Traditionally, teacher education programs have been criticized for inadequately preparing teachers for the complexities of the classroom [19], which applies equally to ECPE teacher education. The theoretical concepts related to ECPE that early childhood teachers learn are often disconnected from the practical realities that they face. This disconnect can result in a lack of confidence, efficacy, and preparedness among new teachers regarding teaching ECPE [7,18]. To address this issue, researchers and educators have explored various strategies and approaches with the aim of bridging the gap.

Research aimed at bridging the gap between theory and practice in ECPE teacher education has focused primarily on several key topics. First, studies have trialed various pedagogical strategies that integrate theory and practice, such as the promotion of reflection, microteaching, and portfolio assessment, which were investigated to facilitate the effective utilization of ECPE theories in practical settings [17,20,21]. Second, topics such as collaborative partnerships, mentoring, and communities of practice have been researched. Collaborative partnerships between ECPE teacher educators and practitioners, as well as the establishment of communities of practice and mentoring programs related to ECPE, have been explored as potential ways of minimizing the discrepancy between theory and practice [22–24]. Finally, research on teachers has explored how they can improve their classroom approaches. Action research, reflective inquiry, practitioner research, and self-study research relating to ECPE have been employed to promote practice-oriented ECPE teacher education [20,25].

As noted above, existing studies have explored strategies and methods to support pre-service teachers in the effective application of the knowledge they have learned in the field. Furthermore, they focus on creating knowledge suitable for in-service teachers. In addition, a common assertion among these studies is the prioritization of incorporating the integrated voices of theorists and practitioners as an effective means to minimize the gap between theory and practice in ECPE teacher education. As its primary approach, therefore, this study adopted the Delphi research method, which enables the integration of ECPE theorist and practitioner perspectives, thus synthesizing diverse viewpoints from theory and practice to derive consensus and provide evidence-based guidelines or strategies for more practice-oriented teacher education programs. This method was selected based on the

belief that it would facilitate the development of contextually relevant teacher education programs by integrating theory and practice based on the voices of various stakeholders.

In line with these considerations, this study aims to explore effective teacher education strategies to normalize physical education for preschool-aged children (aged 2–6 years) in Korea. By leveraging the expertise of both scholars and practitioners, this study seeks to identify practical and evidence-based teacher education strategies that integrate theory and practice, prioritize children's health and growth, and address the specific challenges faced in an ECPE context. This study is expected to contribute to the continued sports participation of young children, which serves as the foundation for lifelong engagement in sports. It does so by providing goals, content, methods, and evaluation criteria for both pre-service and in-service teacher education, aimed at bridging the gap between theory and practice in ECPE. In essence, the results of this study hold significance as they address the theory–practice divide, offering avenues for collaboration between theorists and practitioners to resolve the challenges and difficulties faced in ECPE. This is essential for the sustainable development of physical and sports culture.

## 2. Methods

### 2.1. Research Design

This study aims to derive ECPE teacher education strategies that minimize the gap between theory and practice to normalize ECPE in Korea. To achieve this objective, the Delphi research method was adopted as the primary research approach. This method synthesizes the opinions and judgments of a group of experts to derive a consensus on educational goals, directions, and other related matters [26] and was found to be particularly useful in this study in eliciting consensus between researchers and practitioners for the development of directions and strategies in bridging the gap between theory and practice in ECPE. Three rounds of Delphi surveys were conducted with scholars and practitioners in ECPE for the purpose of applying their expertise to the development of teacher education strategies that integrate theory and practice effectively.

### 2.2. Participant Eligibility and Panel Recruitment

A total of 20 experts were selected for the Delphi survey, including 9 researchers in the field of ECPE and 11 practitioners related to ECPE (aged 2–6 years) in Korea. The group of ECPE researchers consisted of individuals with more than 5 years of experience in ECPE-related education and research, including professors ($n = 3$), university lecturers ($n = 2$), and researchers ($n = 4$). The group of ECPE practitioners comprised teachers ($n = 7$) who showed a strong interest in ECPE and actively engaged in classroom research and development, as well as teachers ($n = 4$) working as physical education instructors in ECPE. Similar to the group of ECPE researchers, the group of practitioners also included individuals with more than 5 years of teaching experience.

### 2.3. Data Collection

The data were collected through a literature review, Delphi surveys, and expert meetings. First, a literature review was conducted to comprehensively review previous studies focusing on keywords such as "teacher education", "early childhood teacher education", and "ECPE teacher education" to obtain specific ideas for exploring teacher education strategies that integrate theory and practice.

Next, the Delphi survey was conducted in three rounds. The first Delphi survey gathered expert opinions on strategies for pre-service and in-service ECPE teachers through open-ended questionnaires. The open-ended questionnaire presented the concept and characteristics of pre-service teacher education and teaching strategies identified based on the review of previous studies. The second Delphi survey consisted of a structured closed-ended questionnaire to evaluate the suitability of the results from the first survey. Additionally, experts were asked to provide further opinions on teacher education strategies that integrate theory and practice. The third Delphi survey presented the response

results from the second survey and the individual responses of each expert to evaluate the suitability of each item and derive a final consensus.

Finally, expert meetings were conducted with five ECPE experts (three professors and two teachers) in three sessions. In the first expert meeting, the results of the literature review on teacher education strategies were reviewed, and the direction for constructing the Delphi survey was discussed. The second and third expert meetings were held during the data analysis process of the Delphi survey to ensure the objectivity and reliability of the data. The inductive category analysis process from the first Delphi survey and the verification of terminology and concept definitions in the second and third Delphi surveys were examined.

### 2.4. Data Analysis

The Delphi survey results were analyzed by assigning codes to the questionnaire items, organizing the data in MS Excel, and analyzing it using the SPSS 22.0 software. Since the first Delphi survey was conducted with open-ended questionnaires, inductive content analysis was performed to categorize and derive the categories and sub-strategies of pre-service and in-service teacher education strategies. The panel was requested to provide diverse statements regarding the development of pre-service teacher education approaches for fostering competency in ECPE teaching. Based on the strategies suggested by the panelists, the meaning was conceptualized by extracting key terms. Then, categorization was performed by integrating and deleting overlapping elements. During the categorization process, the nature of the instructional strategies identified in the analysis of prior studies was considered. The interrelationships between the strategies were repeatedly reviewed to derive higher-level categories and specify sub-elements within them for teacher education.

The second and third Delphi surveys, which were conducted with a closed-ended questionnaire, involved deriving the mean, standard deviation, and median for each item. Content validity was calculated to determine the agreement level in the research content. Content validity is a key criterion for item selection in the Delphi survey, and the criteria vary depending on the number of experts participating in the survey [27]. When the number of experts is 10, the criterion is set at 0.62; for 20 experts, it is 0.42; and for 25 experts, it is 0.37 [28]. Given that this study involved 20 experts as the Delphi survey panel, only items within the range of 0.42 or above were considered in the final research results.

## 3. Results

The teacher education strategies for enhancing competence in early childhood physical education teaching aimed at minimizing the gap between theory and practice were presented, with a distinction drawn between those aimed at pre-service and in-service teachers.

### 3.1. Pre-Service Teacher Education Strategies for ECPE

3.1.1. First Delphi Survey

The first Delphi survey was conducted with open-ended questionnaires targeting Delphi panels to derive pre-service teacher education strategies for improving teaching competency in the field of physical education for young children. The results of the inductive content analysis for the open-ended questionnaires; the five categories of goals, content, methods, evaluation, and environmental aspects; and 20 pre-service teacher education strategies were structured (see Table 1).

**Table 1.** First Delphi survey results on pre-service teacher education strategies for ECPE.

| Category | Summary of Open-Ended Survey Responses |
|---|---|
| Goals | Education focused on practical competency, education as a joyful experience, education for fostering healthy exercise habits, and education centered on early childhood/physical play |
| Content | Education on movement skills and fitness components, education related to fine motor development, education on physical activities according to development, and education on various types of physical activities |
| Methods | Play-centered lessons, observational learning, microteaching, utilization of various teaching methods, and collaborative learning with education experts |
| Evaluation | Composition of class portfolio, evaluation by education experts and field teachers, and peer evaluation |
| Environment | Mandatory completion of physical education courses, specialization/diversification/integration curriculum, enhanced field practicum, and improvement of employment system |

### 3.1.2. Second Delphi Survey

Based on the results of the first Delphi survey on pre-service teacher education strategies for enhancing competency in ECPE, the second Delphi survey was conducted to assess the suitability of each item. The derived values for suitability assessment included the mean, standard deviation, median, and content validity rate. The results are presented in Table 2. In the second Delphi survey, the panelists' opinions on the pre-service teacher education strategies showed an average of 4.0 or higher for all detailed strategies, except for the utilization of various teaching methods ($M = 3.75$, $SD = 0.77$) in the methods category.

**Table 2.** Second Delphi survey results on pre-service teacher education strategies for ECPE.

| No | Category | Detailed Strategies | *M* | *SD* | Median | CVR |
|---|---|---|---|---|---|---|
| 1 | Goals | Education focused on practical competency | 4.10 | 0.94 | 4.00 | 0.60 |
| 2 | | Education as a joyful experience | 4.30 | 0.78 | 4.50 | 0.60 |
| 3 | | Education for fostering healthy exercise habits | 4.10 | 0.77 | 4.00 | 0.50 |
| 4 | | Education centered on early childhood/physical play | 4.40 | 0.58 | 4.00 | 0.90 |
| 5 | Content | Education on movement skills and fitness components | 4.25 | 0.62 | 4.00 | 0.80 |
| 6 | | Education related to fine motor development | 4.55 | 0.59 | 5.00 | 0.90 |
| 7 | | Education on physical activities according to development | 4.50 | 0.67 | 5.00 | 0.80 |
| 8 | | Education on various types of physical activities | 4.50 | 0.59 | 5.00 | 0.90 |
| 9 | Methods | Play-centered lessons | 4.75 | 0.43 | 5.00 | 1.00 |
| 10 | | Observational learning | 4.55 | 0.59 | 5.00 | 0.90 |
| 11 | | Microteaching | 4.20 | 0.98 | 4.00 | 0.70 |
| 12 | | Utilization of various teaching methods | 3.75 | 0.77 | 4.00 | 0.30 |
| 13 | | Collaborative learning with education experts | 4.65 | 0.57 | 5.00 | 0.90 |
| 14 | Evaluation | Composition of class portfolio | 4.10 | 0.70 | 4.00 | 0.60 |
| 15 | | Evaluation by education experts and field teachers | 4.30 | 0.64 | 4.00 | 0.80 |
| 16 | | Peer evaluation | 4.20 | 0.58 | 4.00 | 0.60 |
| 17 | Environment | Mandatory completion of physical education courses | 4.55 | 0.67 | 5.00 | 0.80 |
| 18 | | Specialization/diversification/integration curriculum | 4.40 | 0.58 | 4.00 | 0.90 |
| 19 | | Enhanced field practicum | 4.40 | 0.80 | 5.00 | 0.80 |
| 20 | | Improvement of the employment system | 4.05 | 0.80 | 4.00 | 0.40 |

Furthermore, when examining the content validity—a significant criterion for adoption in this study—all detailed strategies showed agreement rates of 0.50 or higher, except for the utilization of various teaching methods (CVR = 0.30) in the methods category and the improvement of the employment system (CVR = 0.40) in the environment category. Particularly, the play-centered lessons in the methods category exhibited a high content validity of 1.00, indicating a strong consensus among experts regarding the effectiveness of play-centered lessons.

### 3.1.3. Third Delphi Survey

In the second Delphi survey, although the panelists reached some consensus, additional feedback and opinions were considered. As a result, self-assessment and a qualification system were added as detailed strategies in the evaluations and environmental categories, respectively. A third Delphi survey was conducted to revalidate the appropriateness of the revised and supplemented pre-service teacher education strategies in physical education for young children. As shown in Table 3, the average values and CVRs for most of the detailed strategies in the third survey remained the same or slightly increased compared with the second survey. However, the utilization of various teaching methods, which had a low consensus rate of 0.30 in the second survey, was finally removed, as it showed a CVR of 0.40 in the third survey. On the other hand, the improvement of the employment system, which had a consensus rate of 0.40 in the second survey, was adopted because it showed a CVR of 0.60 in the third survey. Additionally, the two newly added items in the third survey had a consensus rate of 0.70 for self-evaluation, indicating its final adoption, while the introduction of a qualification system had a consensus rate of 0.00, leading to its removal.

**Table 3.** Third Delphi survey results on pre-service teacher education strategies for ECPE.

| No | Category | Detailed Strategies | *M* | *SD* | Median | CVR | Note |
|----|----------|---------------------|-----|------|--------|-----|------|
| 1 | Goals | Education focused on practical competency | 4.25 | 0.70 | 4.00 | 0.70 | |
| 2 | | Education as a joyful experience | 4.50 | 0.59 | 5.00 | 0.90 | |
| 3 | | Education for fostering healthy exercise habits | 4.50 | 0.50 | 4.50 | 1.00 | |
| 4 | | Education centered on early childhood/physical play | 4.55 | 0.59 | 5.00 | 0.90 | |
| 5 | Content | Education on movement skills and fitness components | 4.45 | 0.50 | 4.00 | 0.80 | |
| 6 | | Education related to fine motor development | 4.75 | 0.54 | 5.00 | 1.00 | |
| 7 | | Education on physical activities according to development | 4.60 | 0.58 | 5.00 | 0.90 | |
| 8 | | Education on various types of physical activities | 4.55 | 0.59 | 5.00 | 1.00 | |
| 9 | Methods | Play-centered lessons | 4.80 | 0.40 | 5.00 | 1.00 | |
| 10 | | Observational learning | 4.60 | 0.49 | 5.00 | 0.90 | |
| 11 | | Microteaching | 4.25 | 0.77 | 4.00 | 0.70 | |
| 12 | | Utilization of various teaching methods | 3.90 | 1.09 | 4.00 | 0.40 | Final deleted |
| 13 | | Collaborative learning with education experts | 4.65 | 0.57 | 5.00 | 0.90 | |
| 14 | Evaluation | Composition of class portfolio | 4.20 | 0.87 | 4.00 | 0.70 | |
| 15 | | Evaluation by education experts and field teachers | 4.40 | 0.81 | 4.00 | 0.90 | |
| 16 | | Peer evaluation | 4.35 | 0.57 | 4.00 | 0.90 | |
| Added | | Self-evaluation | 4.35 | 0.85 | 5.00 | 0.70 | Final adoption |
| 17 | Environment | Mandatory completion of physical education courses | 4.60 | 0.58 | 5.00 | 0.80 | |
| 18 | | Specialization/diversification/integration curriculum | 4.50 | 0.50 | 4.50 | 0.90 | |
| 19 | | Enhanced field practicum | 4.65 | 0.57 | 5.00 | 0.80 | |
| 20 | | Improvement of employment system | 4.05 | 0.80 | 4.00 | 0.60 | Final adoption |
| Added | | Introduction of a qualification system | 3.75 | 1.04 | 3.50 | 0.00 | Final deleted |

Based on the three rounds of Delphi surveys, the pre-service teacher education strategies for fostering ECPE teaching competency were categorized into five main categories and twenty detailed strategies. The final results are presented in Table 4.

**Table 4.** Final Delphi survey results on pre-service teacher education strategies for ECPE.

| No | Category | Detailed Strategies | Explanations |
|----|----------|---------------------|--------------|
| 1 | Goals | Education focused on practical competency | A strategy aimed at cultivating practical competency (knowledge, skills, and attitudes) for effective implementation of ECPE in the field. |
| 2 | | Education as a joyful experience | A strategy to enhance the perception of ECPE as enjoyable, fostering interest and self-confidence. |
| 3 | | Education for fostering healthy exercise habits | A strategy to cultivate healthy exercise habits through participation in ECPE-related education. |
| 4 | | Education centered on early childhood/physical play | A strategy implemented based on understanding the importance of a child/play-centered approach in ECPE. |
| 5 | Content | Education on movement skills and fitness components | A strategy for learning not only body expression but also fundamental movement skills and aspects related to physical fitness. |
| 6 | | Education related to fine motor development | A strategy for learning content related to everyday play that promotes not only gross motor activities (such as throwing a ball using an arm) but also fine motor development (such as picking up a small ball with fingers). |
| 7 | | Education on physical activities according to development | A strategy for learning appropriate physical activity content based on the developmental status of young children. |
| 8 | | Education on various types of physical activities | A strategy to learn various physical activity contents with high field utilization |
| 9 | Methods | Play-centered lessons | A strategy for incorporating play-centered approaches into ECPE classes to ensure practical applications in real-life settings. |
| 10 | | Observational learning | A strategy for enhancing understanding of actual situations in ECPE by observing exemplary classes. |
| 11 | | Microteaching | A strategy for improving teaching abilities in ECPE by conducting small-scale mock classes. |
| 12 | | Collaborative learning with education experts | A strategy for providing more professional and practical feedback through collaborative lessons with education experts and practicing teachers. |
| 13 | Evaluation | Composition of class portfolio | A strategy for documenting and utilizing what has been learned in ECPE classes as a knowledge repository. |
| 14 | | Evaluation by education experts and field teachers | A strategy for promoting practical competency through professional and practical evaluations by educational experts and on-site teachers. |
| 15 | | Peer evaluation | A strategy to help to grow together by reflecting evaluations of pre-service teachers who participated in ECPE classes. |
| 16 | | Self-evaluation | A strategy for facilitating personal growth by promoting self-reflection on one's ECPE class activities based on reflective thinking. |
| 17 | Environment | Mandatory completion of physical education courses | A strategy for transforming ECPE-related subjects from elective courses to mandatory courses and ensuring the compulsory completion of a certain number of credits. |
| 18 | | Specialization/diversification/integration curriculum | A strategy for enhancing knowledge acquisition in ECPE by refining and diversifying related subjects and promoting systematic organization through hierarchical or interconnected relationships among the subjects, enabling continuous learning. |
| 19 | | Enhanced field practicum | A strategy to expand the duration of field practica and make practical experience in ECPE mandatory, enabling the application of practical skills. |
| 20 | | Improvement of employment system | A strategy to increase the weightage of ECPE in teacher certification exams, encouraging the interest of pre-service teachers in early childhood physical activity education and bringing about changes in teacher education. |

*3.2. In-Service Teacher Education Strategies for ECPE*

3.2.1. First Delphi Survey

The first Delphi survey was conducted to derive strategies aimed at enhancing the professional development of in-service teachers in ECPE. Open-ended questionnaires were administered to Delphi panelists, presenting the concepts and characteristics of in-

service teacher education and instructional strategies. The results of the inductive content analysis for open-ended questionnaires; the five categories of goals, content, methods, evaluation, and environmental aspects; and sixteen in-service teacher education strategies were structured (see Table 5).

**Table 5.** First Delphi survey results on in-service teacher education strategies for ECPE.

| Category | Summary of Open-Ended Survey Responses |
|---|---|
| Goals | Education on the importance of physical activity for young children and education on understanding the value of physical play for young children |
| Content | Physical activity and sports skills, education on order and safety, and digital utilization training |
| Methods | Sharing of programs and teaching materials, collaboration between education experts and practicing teachers, horizontal teacher education, case-based teacher education, and building a learning community for teachers |
| Evaluation | Application of bonus point system and diagnosis and supplementation of leadership competency |
| Environment | Compulsory child physical activity teacher training, diversification of early childhood physical activity teacher training and provision of information, active support for teacher learning communities, and securement of places for early childhood physical activity teacher training |

### 3.2.2. Second Delphi Survey

Based on the results of the first Delphi survey on in-service teacher education strategies for enhancing competency in ECPE, the second Delphi survey assessed the suitability of each item. The suitability assessment involved determining the mean, standard deviation, median, and content validity rate. The results are presented in Table 6. In the second Delphi survey, the panelists' opinions on strategies for ECPE teacher training showed that, except for digital utilization training ($M = 3.85$, $SD = 0.79$) in the content category and the application of a bonus points system ($M = 3.65$, $SD = 0.90$) in the evaluation category, all other detailed strategies received an average rating of 4.0 or higher.

**Table 6.** Second Delphi survey results on in-service teacher education strategies for ECPE.

| No | Category | Detailed Strategies | M | SD | Median | CVR |
|---|---|---|---|---|---|---|
| 1 | Goals | Education on the importance of physical activity for young children | 4.40 | 0.58 | 4.00 | 0.90 |
| 2 | | Education on understanding the value of physical play for young children | 4.45 | 0.59 | 4.50 | 0.90 |
| 3 | Content | Physical activity and sports skills | 4.50 | 0.59 | 5.00 | 0.90 |
| 4 | | Education on order and safety | 4.55 | 0.67 | 5.00 | 0.80 |
| 5 | | Digital utilization training | 3.85 | 0.79 | 4.00 | 0.30 |
| 6 | Methods | Sharing of programs and teaching materials | 4.65 | 0.48 | 5.00 | 1.00 |
| 7 | | Collaboration between education experts and practicing teachers | 4.60 | 0.58 | 5.00 | 0.90 |
| 8 | | Horizontal teacher education | 4.50 | 0.59 | 5.00 | 1.00 |
| 9 | | Case-based teacher education | 4.55 | 0.50 | 5.00 | 1.00 |
| 10 | | Building a learning community for teachers | 4.40 | 0.58 | 4.00 | 0.90 |
| 11 | Evaluation | Application of bonus points system | 3.65 | 0.90 | 4.50 | 0.30 |
| 12 | | Diagnosis and supplementation of leadership competency | 4.45 | 0.59 | 4.00 | 0.90 |
| 13 | Environment | Compulsory child physical activity teacher training | 4.00 | 0.77 | 5.00 | 0.60 |
| 14 | | Diversification of early childhood physical activity teacher training and provision of information | 4.50 | 0.59 | 5.00 | 0.90 |
| 15 | | Active support for teacher learning community | 4.65 | 0.57 | 5.00 | 0.90 |
| 16 | | Securement of places for early childhood physical activity teacher training | 4.40 | 0.66 | 4.50 | 0.80 |

In addition, when examining the content validity—a significant criterion for adoption in this study—all detailed strategies showed agreement rates of 0.80 or higher, except for digital utilization training (CVR = 0.30) in the content category and the application of a bonus points system (CVR = 0.30) in the evaluation category. Particularly, in the method category, the sharing of programs and teaching materials, horizontal teacher education, and case-based teacher education demonstrated a high content validity of 1.00, indicating a strong consensus among experts regarding the detailed strategies for in-service teacher education in ECPE.

### 3.2.3. Third Delphi Survey

In the second Delphi survey, although the panelists reached a degree of consensus, additional suggestions from the experts' other opinions were incorporated. As a result, in the evaluation category, the monitoring of early childhood physical activity classes, and in the environment category, early childhood physical activity education and conducting competitions were added as detailed strategies. A third Delphi survey was conducted to revalidate the appropriateness of the revised and improved strategies for in-service teacher education in early childhood physical activity. As shown in Table 7, compared with the second survey, most of the mean values and CVRs for the detailed strategies in the third survey either remained the same or slightly increased. However, the low CVRs of 0.40 for digital utilization training in the content category and the application of a bonus points system in the evaluation category in the second survey led to their final exclusion in the third survey. The consensus rate for the added items in the third survey was 0.50 for monitoring early childhood physical activity classes, which was ultimately adopted, while it was 0.30 for early childhood physical activity education and conducting competitions, resulting in their final exclusion. Additionally, in the environment category, while the mean for the mandatory training of early childhood physical activity teachers increased from 4.00 to 4.15, the consensus rate decreased from 0.60 to 0.40, leading to its final exclusion.

**Table 7.** Third Delphi survey results on in-service teacher education strategies for ECPE.

| No | Category | Detailed Strategies | *M* | *SD* | Median | CVR | Note |
|---|---|---|---|---|---|---|---|
| 1 | Goal | Education on the importance of physical activity for young children | 4.60 | 0.49 | 5.00 | 1.00 | |
| 2 | | Education on understanding the value of physical play for young children | 4.55 | 0.59 | 5.00 | 0.90 | |
| 3 | Content | Physical activity and sports skills | 4.55 | 0.50 | 5.00 | 0.90 | |
| 4 | | Education on order and safety | 4.75 | 0.43 | 5.00 | 1.00 | |
| 5 | | Digital utilization training | 3.90 | 0.77 | 4.00 | 0.40 | Final deleted |
| 6 | Method | Sharing of programs and teaching materials | 4.65 | 0.48 | 5.00 | 0.90 | |
| 7 | | Collaboration between education experts and practicing teachers | 4.60 | 0.49 | 5.00 | 0.90 | |
| 8 | | Horizontal teacher education | 4.60 | 0.49 | 5.00 | 1.00 | |
| 9 | | Case-based teacher education | 4.70 | 0.46 | 5.00 | 1.00 | |
| 10 | | Building a learning community for teachers | 4.45 | 0.59 | 4.50 | 0.90 | |
| 11 | Evaluation | Application of bonus points system | 3.85 | 0.96 | 4.00 | 0.40 | Final deleted |
| 12 | | Diagnosis and supplementation of leadership competency | 4.50 | 0.59 | 5.00 | 0.90 | |
| Added | | Monitoring early childhood physical activity classes | 4.00 | 0.71 | 4.00 | 0.50 | Final adoption |

**Table 7.** *Cont.*

| No | Category | Detailed Strategies | *M* | *SD* | Median | CVR | Note |
|---|---|---|---|---|---|---|---|
| 13 | | Compulsory child physical activity teacher training | 4.15 | 0.79 | 4.00 | 0.40 | Final deleted |
| 14 | | Diversification of early childhood physical activity teacher training and provision of information | 4.60 | 0.49 | 5.00 | 0.90 | |
| 15 | Environment | Active support for teacher learning community | 4.75 | 0.43 | 5.00 | 0.90 | |
| 16 | | Securement of places for early childhood physical activity teacher training | 4.50 | 0.59 | 5.00 | 0.90 | |
| Added | | Early childhood physical activity education and conducting competitions | 3.90 | 0.89 | 4.00 | 0.30 | Final deleted |

Based on the Delphi survey conducted over three rounds, the strategies for enhancing ECPE teaching competency were categorized into five main categories and fourteen detailed strategies. The final results are presented in Table 8.

**Table 8.** Final Delphi survey results on in-service teacher education strategies for ECPE.

| No | Category | Detailed Strategies | Explanations |
|---|---|---|---|
| 1 | Goal | Education on the importance of physical activity for young children | The strategy aims to remind students of the importance and enjoyment of ECPE, as well as to cultivate interest and commitment to its implementation. |
| 2 | | Education on understanding the value of physical play for young children | The strategy aims to promote an understanding of the educational value of play in ECPE and support the shift from a teacher-led to a child-centered approach in implementing ECPE. |
| 3 | Content | Physical activity and sports skills | The strategy aims to cover a wide range of physical activities encompassed in ECPE and to help develop sufficient practical skills to teach these activities effectively. |
| 4 | | Education on order and safety | The strategy involves providing education on safety measures and accident prevention techniques in ECPE classes, as well as teaching methods to effectively handle safety incidents. |
| 5 | | Sharing of programs and teaching materials | The strategy focuses on sharing various teaching programs and instructional materials (such as videos, music, teaching aids, etc.) that can be used in ECPE classes. |
| 6 | | Collaboration between education experts and practicing teachers | The strategy aims to facilitate professional and practical support for educators through collaboration with educational experts and practicing teachers. |
| 7 | Method | Horizontal teacher education | The strategy involves avoiding traditional content delivery methods and, instead, conducting classes through workshops, seminars, discussions, and other interactive approaches. |
| 8 | | Case-based teacher education | The strategy involves presenting exemplary and unsuccessful cases of ECPE to provide direct and indirect experiences of effective teaching methods in early childhood physical activities. |
| 9 | | Building a learning community for teachers | The strategy aims to establish a professional learning community for ECPE, fostering ongoing sharing and learning among educators. |

**Table 8.** *Cont.*

| No | Category | Detailed Strategies | Explanations |
|---|---|---|---|
| 10 | Evaluation | Diagnosis and supplementation of leadership competency | The strategy involves utilizing photos, videos, assessment tools, and other resources related to ECPE to assess and improve current teaching competencies in this domain. |
| 11 | | Monitoring early childhood physical activity classes | The strategy involves continuously observing early childhood teachers during physical education classes to enhance their instructional competencies. |
| 12 | Environment | Diversification of early childhood physical activity teacher training and provision of information | The strategy aims to expand opportunities for professional development in ECPE by providing various types of teacher training programs and establishing an efficient and systematic system for managing related information. |
| 13 | | Active support for teacher learning community | The strategy aims to promote the activation of a learning community for ECPE teachers by providing time, financial resources, and administrative assistance. |
| 14 | | Securement of places for early childhood physical activity teacher training | The strategy aims to secure adequate space for the efficient and ongoing implementation of teacher training related to ECPE. |

## 4. Discussion

Based on the results of this study, we highlight the educational implications for ECPE teacher education that minimizes the gap between theory and practice. These implications will be discussed separately for pre-service and in-service teacher education strategies.

### 4.1. Educational Implications for Pre-Service ECPE Teacher Education Strategies

First, a shift in the goals and content of teacher education is required, with a focus on developing not only theoretical knowledge but also "practical competency" that allows teachers to adapt flexibly to the diverse environments of educational settings. It is necessary to move beyond theoretical understanding and cultivate the ability to apply appropriate teaching content and strategies in real situations for the purpose of interacting with students and providing instruction in actual educational contexts [19]. In this sense, an approach that fosters practical competency and prioritizes educational content that facilitates its manifestation will help pre-service teachers apply their theoretical grounding in ECPE to actual educational situations and facilitate effective learning experiences.

Second, various effective strategies have been derived from the method and evaluation categories of pre-service teacher education, including classroom observation; microteaching; the integration of learning and evaluation with educational experts and field teachers; and the construction of teaching portfolios. These strategies will allow the gap between theory and practice to be bridged, as traditional pre-service teacher education often focuses exclusively on cultivating theoretical knowledge without addressing how it can be implemented in classroom settings [19]. The integration of theory and practice in educational methods can provide pre-service teachers with simulated and experiential learning opportunities that closely resemble real-world ECPE classrooms. This can help them to understand and respond to the complex and dynamic situations that arise in instructional settings [18,20,25].

Third, the results indicate that collaboration between experts and field teachers who possess practical knowledge acquired through extensive research and instruction in ECPE is essential for the integration of theory and practice [22–24]. This is particularly the case for pre-service teachers who may lack practical experience, for whom exposure to unpredictable real-life situations through collaboration with educational experts and teachers in the field is crucial. Moreover, given that pre-service teachers may have limited opportunities to learn about the functional aspects of physical activities in early childhood education, the assembly of teaching portfolios that document routines and other pedagogical mate-

rials may serve as valuable resources that support their entry into the field [21]. Using these evaluation methods, pre-service teachers can prepare for and adapt to actual ECPE situations, enhance their professional competence and confidence, and develop the ability to facilitate effective ECPE experiences [25].

Finally, within the environment category of pre-service teacher education, strengthening field practica emerged as particularly important for bridging the gap between theory and practice and creating an environment that offers opportunities to acquire practical experience in real educational settings. Field practica provide pre-service teachers with opportunities to interact with children; experience the dynamic nature of physical education in early childhood; and develop essential skills, such as problem-solving and communication [29]. Additionally, field practica help to foster the necessary expertise and confidence in pre-service teachers by allowing them to apply their theoretical knowledge to real-world situations [29]. However, previous studies have noted that the duration of field practica in early childhood education is often insufficient and that the opportunities to engage in practical activities relating to young children's physical education are limited, even when pre-service teachers are placed in educational settings in Korea [15,16]. Therefore, it is essential to establish a pre-service teacher education environment that actually furnishes opportunities to engage in activities related to physical education for young children.

Alongside field practica, it is crucial to note the derived strategy of "specialization, diversification, and integration" in physical education-related courses, which may reflect the content-poor and fragmented nature of physical education-related courses in teacher education in Korea, in line with previous studies [15,16]. Specialization allows for an in-depth understanding of theoretical background knowledge in physical education and the development of skills that facilitate its application in actual educational situations. Furthermore, the diversification of courses will provide pre-service teachers with opportunities to experience and learn various physical education activities and teaching strategies, which will help them acquire the ability to provide suitable physical education to diverse children. Moreover, the integration of physical education courses with other educational domains contributes to offering an optimal early childhood education experience.

### 4.2. Educational Implications for In-Service ECPE Teacher Education Strategies

First, the importance of physical activities for young children and the value of their physical play should be emphasized during in-service teacher training. This differs from the focus on developing practical competency during pre-service early childhood teacher education. In other words, it is crucial that the perspective of teachers with experience in the field and who understand the value of physical activities and physical play at the goal level be considered. This finding supports previous research indicating that teachers who understand the importance of ECPE are more capable of effectively guiding young children in physical activities [21,30]. Furthermore, it suggests that teachers must apply theoretical directions in practice to ensure that ECPE is effectively implemented.

Second, in the content category of in-service teacher education, the enhancement of physical activity and sports skills to facilitate young children's physical activities emerged as key educational content. The ability to perform physical activities and sports enables teachers to effectively teach young children appropriate movement and body control techniques and provide appropriate support for their physical development [20,25]. Therefore, physical activity and sports skills serve as a means of bridging the gap between theoretical knowledge and educational practice. As such, developing the skills required to effectively teach young children's physical activities is a crucial component of in-service teacher education.

Regarding the method and evaluation categories in the Delphi survey, it is interesting to note that pre-service and in-service teacher education share commonalities and differences in terms of approaches and evaluations that have the potential to narrow the gap between theory and practice. First, one commonality that came to light was the identification of collaboration with experienced educational professionals and practicing teachers

who have rich experience in educational settings as an important strategy in both pre-service and in-service teacher education. This finding suggests the need for someone with practical expertise in ECPE to assist not only pre-service early childhood teachers who lack experience in the field but also in-service teachers with some field experience.

In contrast to pre-service teacher education, in-service teacher education has identified strategies such as mentoring and building learning communities for teachers. These results show that in-service teachers who teach physical activities often experience difficulties in teaching compared with other subjects, so exchange and cooperation with experienced senior teachers or fellow teachers are necessary. In mentoring among in-service teachers, it is important for mentors and mentees to engage in a two-way, mutually collaborative process of sharing knowledge, experiences, and concerns [31]. Moreover, establishing learning communities for teachers through collaborative learning is an important strategy for integrating theory and practice in terms of not only teachers' professional competency but also the cultivation of a practical approach. This indicates the effectiveness of a "professional learning community" that can enhance teaching capabilities by sharing knowledge and creating new knowledge through cooperation between teachers [32].

Furthermore, in-service teacher education strategies—unlike pre-service teacher education strategies—require that programs and instructional materials be shared. This suggests a more urgent demand for practical and useful teaching resources among in-service teachers. Various avenues are currently available—both online and offline, such as regional educational support centers—through which programs and instructional materials may be publicly shared. However, mechanisms for sharing programs and resources specifically designed for implementing ECPE in educational settings are currently lacking [15,16,18], highlighting the need for specialized programs and resource-sharing systems that can bridge theory and practice in the guidance of early childhood physical education.

Finally, the results derived in the environment category of in-service teacher education suggest the need for an in-service teacher education environment that connects theory and practice, including the diversification of teacher training for early childhood physical activity teachers and providing active support for teacher learning communities. In-service teachers' various practical responsibilities create an absolute shortage of time for research aimed at developing their classes [18], which necessitates multifaceted forms of teacher training. Additionally, considering the relative difficulty that in-service teachers encounter in their pursuit of relevant teacher education, active support for teacher learning communities—as derived in the method category of the Delphi survey—should be provided to promote an ECPE environment that integrates theory and practice.

## 5. Conclusions

This study aimed to derive strategies for physical education teacher education for preschool-aged children (aged 2–6 years) that minimize the gap between theory and practice. A three-round Delphi survey was conducted with a total of 20 experts in the field of ECPE, including theorists and practitioners. The strategies for integrating theory and practice in ECPE teacher training were presented in two categories: pre-service teacher training strategies and in-service teacher training strategies. Within each category, five categories were identified: a "goals category", a "content category", a "methods category", an "evaluation category", and an "environment category". Based on the study results, educational implications for minimizing the gap between theory and practice in ECPE teacher education were discussed. The teacher education strategies derived from this study hold significance as they prioritize the integration of theory and practice based on the shared opinions of ECPE scholars and practitioners and provide concrete and practical approaches that can actively contribute to the promotion of ECPE.

ECPE plays a crucial role in introducing young people to sports culture, promoting lifelong sports participation, and fostering healthy habits. The division between theory and practice should not be binary, just as scholars (such as researchers) and practitioners (such as teachers) should maintain a collaborative relationship. Researchers should consolidate

diverse opinions from the field to identify appropriate directions and approaches for ECPE, as well as support teacher education and teacher communities. Meanwhile, practitioners must evolve into educators who produce knowledge and grow within the field. For the sustainability and advancement of ECPE, the connection between theory and practice, achieved through collaboration between researchers and practitioners, is essential. Future studies need to focus on how researchers and practitioners might best collaborate to implement the various strategies proposed herein.

**Author Contributions:** Conceptualization, S.P. and H.J.; methodology, S.P.; validation, H.J.; data curation, S.P. and H.J.; writing—original draft preparation, S.P.; writing—review and editing, H.J. All authors have read and agreed to the published version of the manuscript.

**Funding:** This research was supported by the Ministry of Education of the Republic of Korea and the National Research Foundation of Korea (NRF-2022S1A5B5A16053252).

**Institutional Review Board Statement:** The study was conducted according to the guidelines of the Declaration of Helsinki and was approved by the Institutional Review Board of Gachon University (1044396-202304-HR-060-01).

**Informed Consent Statement:** Informed consent was obtained from all subjects involved in the study.

**Data Availability Statement:** The data are not publicly available because of privacy issues.

**Conflicts of Interest:** The authors declare no conflict of interest. The funders had no role in the design of the study; in the collection, analyses, or interpretation of the data; in the writing of the manuscript; or in the decision to publish the results.

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
