# Peer review of "Bridging the Gap: Integrating Theory and Practice for Early Childhood Physical Education Teacher Education in Korea"

_sustainability, doi:10.3390/su151914397_

Round 1

Reviewer 1 Report

- Intro.

1. This study explored strategies for early childhood physical education teacher education that minimize the gap between theory and practice. And author(s) emphasize(s) the key phrases such as ‘the discrepancy between theory and practice’. However, I don’t see any evidences about what kinds of gaps exist in the physical education classroom in Korea. Please describe the specific evidences reflecting the practical settings.

2. Line 50 -> I think this is not true. It sounds like that the author(s) is(are) disparaging early childhood physical education and teachers in Korea. This statement should be based on the literature. How is(are) the author(s) sure of it?

3. Line 78, 81, & 86 -> Please revise grammatical errors.

4. Line 90 -> Please refer the logical reasons that how or where the two categories (pre-service & in-service teacher training strategies) are from.

 - Results

1. I don’t see how five components (goal, content, method, evaluation, & environment categories) come from. Please describe theoretical background in the Intro.

2. Based on the outcomes shown in the Tables, the Detailed Strategies just look like abstractive concepts. Readers all would like to ask what the ‘education focused on practical competency’ are, or what the ‘education for fostering healthy exercise habits’ are, etc.

3. Thus, in Introduction, most of all, based on the classroom settings in Korea it should be described what the discrepancy exists between theory and practice.

There are some sentences with grammatical error.

Author Response

Please refer the attached file for the table of response to reviewer. 

Reviewer 2 Report

Peer Reviewed Comments:

Introduction

-          The authors continue to refer to the term “early childhood,” but it is suggested early on in the introduction to more clearly define the ages of “early childhood”. I think this would provide a clearer picture in terms of the focus and ages of the children to whom it is referring.

-          Lines 46 – 49 highlight a crucial point, which is the need to receive guidance for quality instruction in early childhood physical education which as the authors say “is crucial”, but is their research to further support this statement? I think it would add to the overall credibility of the statement.

-          The introduction as a whole is obvious and well-written. The others provide a strong picture of the need for the current research study.

-          The references are overall relevant and up to date.

Methods Section

-          The research design is straightforward, and the authors highlight why the Delphi research method was incorporated as their research method.

-          The overall description of research participants was evident.

-          Page 3; Lines 128 – 131 When discussing the early childhood physical education practitioner group, it might be worth identifying the number of participants who fell into the group that actively engaged in classroom research and development and the group that were teachers working as early childhood physical education instructors.

-          Page 4; Lines 149 – 156: In the paragraph, the authors highlight meetings with five early childhood physical education experts. Would it help to highlight whether they were from the research and practitioner groups?

Results Section

-          Lines 178-191: The paragraphs highlight a review of how data was analyzed from the first Delphi survey. I don’t think this is necessary since this was already discussed previously in the methods section. I don’t see this as a big issue, but something to consider. I think discussing the general purpose of the survey is appropriate, but I don’t feel you need to go into great depth.

-           I found it challenging to keep track of information in the results section. The authors move back and forth between the three Delphi surveys and have eight different tables again, making it difficult to keep track of them. A suggestion would be to try and cut down on the tables and stick to crucial information with regard to the main points that summarize the results. Again, I found myself getting very confused re-reading about the Delphi surveys and keeping track of and trying to comprehend all of the different tables.

Discussion

-          The authors provided a clear discussion section and connected past research to the current student. They offer many strong points to highlight further what was found from the results.

-          I like that the authors have emphasized both the pre-service early childhood physical education teacher education and in-service early childhood physical education teacher education.

Conclusion

-          The discussion section is clear and well-written, and I think the authors have done an excellent job with it. I would consider an addition of future recommendations, though. I didn’t notice this in the discussion or conclusion. Based on the results, what would the suggested next steps be to build off the current research?

-          The authors could include a paragraph in the conclusion that emphasizes something like: “From the current research, it is suggested that to more effectively integrate theory and practice for early childhood physical education teacher education…..” 

Author Response

(The authors gave the same response as above.)

Reviewer 3 Report

Thank you for the opportunity to review this paper that deals with important issue of proper physical education in early childhood. This is interesting and hot topic that deserves publication. The DElphi method used was appropriate and I really have only minor comments:

1. Please define what early childhood physical education is.

2. You should give some examples of differences between gross and fine motor activities.

3. Tables should be self explanatory but in your case abbreviations are not explained (e.g. CVR)

The first comment is mandatory, as I have not found the timeframe that early childhood is defined. Are we talking about pre-school or school children. If school, then there is a difference between first, second and third trimester of y 9-year school program.

Otherwise, I think that this is an important issue and that we all could do more in this field, therefore I suggest the publication of this paper after clarification of the abovementioned issues. 

Author Response

(The authors gave the same response as above.)
